# Exploring the Genetic Networks of HLB Tolerance in Citrus: Insights Across Species and Tissues

**DOI:** 10.3390/plants14121792

**Published:** 2025-06-11

**Authors:** Rodrigo Machado, Sebastián Moschen, Gabriela Conti, Sergio A. González, Máximo Rivarola, Claudio Gómez, Horacio Esteban Hopp, Paula Fernández

**Affiliations:** 1Instituto Nacional de Tecnología Agropecuaria, Estación Experimental Agropecuaria Concordia, Concordia 3200, Argentina; gomez.claudio@inta.gob.ar; 2Instituto Nacional de Tecnología Agropecuaria, Estación Experimental Agropecuaria Famaillá, Famaillá 4132, Argentina; sebamoschen@gmail.com; 3Consejo Nacional de Investigaciones Científicas y Técnicas (CONICET), Buenos Aires 1033, Argentina; conti.gabriela@inta.gob.ar (G.C.); rivarola.maximo@inta.gob.ar (M.R.); fernandez.pc@inta.gob.ar (P.F.); 4Instituto de Agrobiotecnología y Biología Molecular, UEDD INTA CONICET, Buenos Aires 1686, Argentina; gonzalez.sergio@inta.gob.ar (S.A.G.); hopp.esteban@inta.gob.ar (H.E.H.)

**Keywords:** HLB, citrus varieties tolerance, gene co-expression network, RNA-seq analysis

## Abstract

Huanglongbing (HLB), caused mainly by *Candidatus* Liberibacter *asiaticus* (CLas), is a devastating disease threatening citrus production worldwide, leading to leaf mottling, fruit deformation, and significant yield losses. This study generated a comprehensive co-expression network analysis using RNA-seq data from 17 public datasets. Weighted gene co-expression network analysis (WGCNA) was applied to identify gene modules associated with citrus species, tissue types, and days post-infection (DPIs). These modules revealed significant enrichment in biological pathways related to stress responses, metabolic reprograming, ribosomal protein synthesis, chloroplast and plastid function, cellular architecture, and intracellular transport. The results offer a molecular framework for understanding HLB pathogenesis and host response. By elucidating module-specific functions and their correlation with species- and tissue-specific responses, this study provides a robust foundation for identifying key genetic targets. These insights facilitate breeding programs focused on developing HLB-tolerant citrus cultivars, contributing to the long-term sustainability and resilience of global citrus production.

## 1. Introduction

Huanglongbing (HLB), also known as citrus greening, is one of the most severe threats to citrus production worldwide, characterized by yellowing leaves, deformed fruit, and tree death [1,2,3]. This disease, originally documented in southern China [4], was later associated with citrus dieback in India during the 1700s [5,6], leading to the hypothesis that HLB first emerged in India before spreading to China [2,3]. In 1929, a similar disorder was observed in South Africa, distinguished by the incomplete color development at the stylar end of affected fruits [3,7]. The disease spread throughout Asia, the Middle East, and Africa up to the 1990s [8]. For decades, HLB was confined to Asian and African countries until it was discovered in the Americas. The disease was discovered in Brazil in 2004 [9] and the following year in Florida, USA [10]. Now affecting more than 53 countries in Asia, the Americas, Africa, Oceania, and the Caribbean [11], HLB continues to cause significant damage worldwide. These new outbreaks were significant not only because of the large losses that followed, but also because they gave new impetus to research into all aspects of HLB. This disease remains a major risk to citrus production worldwide and a worrying threat to growers in Europe, where it is still absent.

The disease reduces fruit production and quality, leading to lower market prices and consumer demand, while increased management costs, including pest control and removal of infected plants, further exacerbate the substantial economic losses—reported in the billions—affecting farmers, suppliers, and the entire citrus supply chain [12]. Despite these challenges, ongoing research aims to develop more effective management practices and resistant citrus varieties to mitigate the impact of HLB.

Most commercial citrus cultivars worldwide are highly susceptible to HLB, which affects various citrus species, including orange, mandarin, lemon, and grapefruit trees, causing severe damage [13]. The widespread monoculture of a few citrus varieties has led to reduced genetic diversity, which facilitates the rapid spread of HLB among the citrus population [14]. Diverse citrus cultivars have been shown to be more tolerant to huanglongbing HLB, such as the trifoliate orange trees and their hybrids, and the citron and its hybrids, like lemons [14,15]. Over time, other HLB-tolerant species have been identified, including the mandarin hybrid ‘Sugar Belle’ [16] and several wild species such as *Citrus ichangensis* ‘2586’, *Citrus latipes*, sour pummelo, and kaffir lime [17,18,19,20]. The ‘Jackson’ grapefruit-like hybrid has shown significant tolerance compared to the susceptible ‘Marsh’ grapefruit [21]. More recently, Australian lime species and hybrids have also been recognized for their tolerance [14,22,23,24,25]. Persian triploid lime (*Citrus latifolia*) exhibits greater tolerance to HLB than diploid varieties due to superior detoxification processes and physiological traits, such as enhanced callose synthesis in response to infection [26]. To effectively control HLB, it is critical to understand the diversity in responses across citrus cultivars and identify varieties with potential tolerance traits.

The global spread of HLB disease is influenced by several interrelated factors, including the role of its insect vector, climate change, and agricultural practices, all of which are crucial to the effective management and control of the disease. HLB is caused by three species of *Candidatus* Liberibacter: *Candidatus* Liberibacter *asiaticus* (CLas), *Candidatus* Liberibacter *africanus* (CLaf), and *Candidatus* Liberibacter *americanus* (CLam). These species are Gram-negative, phloem-colonizing, psyllid-transmitted, fastidious bacteria classified in the order Rhizobiales (class Alphaproteobacteria) [1,27,28,29]. Among them, CLas is the most widespread, transmitted by the Asian citrus psyllid, *Diaphorina citri*, while CLaf is spread by the African citrus psyllid, *Trioza erytreae*, transmitting the pathogen through feeding on infected plants [1,29]. Studies have shown that nearly all life stages of *D. citri* can harbor the pathogen, facilitating widespread transmission [30]. Climate variability also plays a significant role in influencing the population dynamics of *D. citri*, thereby affecting the incidence of HLB. Changes in temperature and rainfall patterns can alter the distribution and lifecycle of the psyllid, potentially leading to an increase in outbreaks [8,31]. The HLB spread rate is further exacerbated by agricultural practices, like moving infected but asymptomatic plants across regions [8].

Once CLas infects the host plant, it significantly disrupts physiological and biochemical processes, exhibiting complex interactions with both plant and insect hosts that enhance vector transmission [28,32]. One of the primary mechanisms involves the abscisic acid (ABA) signaling pathway, which upregulates callose synthase (*CsCalS11*) expression, leading to increased callose production in response to CLas-associated PAMPs like flagellin [29,33]. Elevated ABA levels activate the *CsABI5*-*CsCalS11* module, essential for callose synthesis [33]. However, CLas also suppresses citrus innate immune responses through proteins such as LasP_235_ and Effector 3 [34]. While these proteins initially promote callose deposition as part of the plant’s defense response, CLas can later inhibit or modulate this process. Transmission electron microscopy shows reduced sieve pore diameter in infected leaves, correlating with callose accumulation and impaired phloem function, further aggravating phloem blockage and the disease’s impact on citrus health and productivity [35,36]. Furthermore, CLas secretory protein SDE3 inhibits host autophagy by degrading *CsATG8* proteins, which are crucial for plant immunity, further promoting disease progression and enhancing virulence in citrus and other plants like *Arabidopsis thaliana* and *Nicotiana benthamiana* [37]. CLas infection also causes a decline in photosynthetic efficiency by reducing chlorophyll levels, increasing cell membrane permeability, and altering reactive oxygen species (ROS) scavenging [38]. Different citrus cultivars exhibit varying responses to CLas; for example, rough lemon trees can rejuvenate despite showing symptoms, while sweet orange trees suffer continuous growth inhibition and eventual dieback [39]. Rough lemon appears to maintain better phloem transport and higher expression of defense-related genes compared to sweet orange, suggesting that robust phloem transport is key to HLB tolerance [39,40].

In recent years, numerous publicly available RNA-seq datasets related to HLB have emerged (e.g., [14,41,42,43]). These studies provide a unique opportunity to explore the molecular basis of citrus responses to CLas infection by re-analyzing and integrating these valuable resources.

Weighted gene co-expression network analysis (WGCNA) is a powerful computational tool used to study complex biological systems, such as plant–pathogen interactions [44]. WGCNA uncovers intricate relationships between genes across tissues and plant varieties, providing a robust framework for unraveling the genetic mechanisms underlying these interactions [45,46]. This method allows researchers to identify co-expressed gene modules that correlate with specific traits, such as disease resistance. In addition, WGCNA can reveal tissue-specific gene expression patterns, which are critical for understanding how the different parts of a plant respond to a pathogen [47]. For example, WGCNA identified modules associated with resistance pathways in *Amorphophallus*, highlighting the role of specific genes in different tissues during pathogen infection [47]. Similarly, in wheat infected by *Puccinia striiformis*, WGCNA highlighted core defense-related genes, such as heat shock proteins and protein kinases, involved in plant defense mechanisms [45]. Recently, ref. [48] showed that hub genes were positively correlated with CLas infection in *Citrus sinensis*, including key genes such as Leaf rust 10 disease-resistance locus receptor-like protein kinase-like, *Ethylene Response Factor 9*, and *TrxR1*, providing insights into the transcriptional regulation during CLas infection.

In this study, we aimed to apply WGCNA to explore gene co-expression networks across multiple tissue types (e.g., leaf and root) and citrus cultivars in a time dependent manner to identify key regulatory genes and pathways involved in the plant response to HLB infection. By integrating RNA-seq data from different sources, we aim to uncover tissue-specific gene expression patterns and pathways perturbed by CLas, ultimately contributing to a deeper understanding of the molecular mechanisms underlying citrus resilience or susceptibility to HLB. This research provides an insight into gene co-expression networks across a wide range of citrus species, tissues, and infection stages, setting the stage for future studies and potential breeding strategies targeting HLB resistance

## 2. Results

### 2.1. Evaluation of RNA-Seq Aligners and Selection of Citrus Reference Genome for Optimal Mapping Efficiency

We analyzed RNA-seq data from 293 samples across five Citrus species and nine tissue types (Appendix A). To assess mapping efficiency, we compared two widely used RNA-seq aligners, STAR and HISAT2, using both the *Citrus sinensis* DHSO v3.0 and *Citrus clementina* v1.0 reference genomes. STAR consistently achieved higher mapping rates across all samples and genomes, with a median of 88.88% (±4.57) uniquely mapped reads for *C. sinensis* DHSO v3.0, compared to a median of 78.06% (±5.77) for HISAT2 on the same genome (Table 1, Appendix A). Similar trends were observed for the *C. clementina* v1.0 genome, where STAR also outperformed HISAT2 (median 89.75% vs. 80.87%).

These findings align with the distinct approaches of the two aligners. STAR uses a maximum mappable seed search combined with suffix arrays, allowing it to rapidly achieve high mapping efficiency, particularly for large datasets [49]. On the other hand, HISAT2 employs a hierarchical indexing strategy and a graph-based approach, which, while efficient for spliced read alignment, resulted in slightly lower mapping efficiency in our dataset [50]. Despite HISAT2’s strengths in handling spliced reads, STAR’s superior speed and sensitivity made it the preferred aligner for our study.

Although both genomes provided similar mapping efficiencies, we selected the C. sinensis DHSO v3.0 for all downstream analyses due to its improved completeness and annotation quality. This choice contributed to a more reliable dataset for downstream gene expression and co-expression analyses, ensuring higher statistical power and biological accuracy.

### 2.2. Quality Control and Data Retention for Further Analyses

Following post-alignment quality control, 230 RNA-seq samples with alignment rates above the established 60% cutoff were retained for downstream analyses. These samples, mapped to *C. sinensis* DHSO v3.0 using STAR, demonstrated a median alignment rate of 88.88% (±4.57). The remaining samples, which showed alignment rates below 60%, were excluded to maintain data reliability for subsequent differential expression and co-expression studies (Figure 1).

### 2.3. Gene Expression Profiling and Principal Component Analysis

Gene expression levels were quantified at the transcript level, resulting in an initial set of 49,567 transcripts. After filtering out low-expressed genes (those with zero variance or missing values), a final set of 32,401 transcripts were retained for further analysis. To ensure comparability across samples and reduce technical noise, the count data were normalized with variance-stabilizing transformation (VST) to stabilize variance, enabling reliable downstream analyses. Furthermore, to remove unwanted batch effects associated with Bioprojects, ComBat batch removal was performed. Principal Component Analysis (PCA) was conducted to explore the global structure of the dataset and identify major sources of variation. The first three principal components explained a cumulative 32.8% of the variance, with PC1 accounting for 17.1%, PC2 for 9.9%, and PC3 for 5.8% (Figure 2A,B). Tissue type was the dominant factor structuring the samples, as seen in the distinct clustering by tissue in both PC1 vs. PC2 and PC2 vs. PC3 plots.

While *Tissue* type was the main driver of variation, as shown by the clustering in the PCA plots (Figure 2), the overall percentage of variance explained by the first principal components remained moderate. This suggests that additional sources of variation are likely present in the dataset. Minimal clustering by *Citrus species* was observed (Appendix A), suggesting that the species-specific genetic background has a weaker influence on global gene expression patterns compared to tissue type. The remaining dispersion in the PCA plots may also reflect uncontrolled biological or technical factors, such as differences in genotype, developmental stage, or sequencing protocols between Bioprojects (Appendix A).

### 2.4. Clustering of Gene Expression Samples

To further investigate the clustering of gene expression profiles and to identify potential outliers, Pearson correlation was used as a distance metric. The resulting dendrogram, shown in Appendix A, revealed clear sample groupings. A height cutoff of 0.5 was applied to exclude 33 outlier samples that likely reflected technical variability, retaining 197 high-quality samples for subsequent analyses. This step enhanced the reliability of the dataset, reducing the influence of outliers on differential expression and co-expression network analyses.

### 2.5. Weighted Co-Expression Network Construction and Module Trait Analysis

The normalized expression values were used to build the weighted gene co-expression network using the ‘WGCNA’ package. A soft-thresholding power of β = 7 was chosen to achieve a scale-free topology, with an R^2^ value of 0.967 (Figure 3A,B). This value balanced network connectivity with biological interpretability. At this threshold, 52 initial modules were identified (Appendix A). Modules with eigengene correlations > 0.75 were merged, resulting in 47 final modules (Figure 3C; Appendix A).

The distribution of genes per module is detailed in Appendix A, with key modules including *turquoise* (17.10%, 5539 genes), *blue* (12.43%, 4029 genes), and *brown* (7.12%, 2306 genes). A total of 552 genes (1.70%) remained unclustered, forming the *grey60* module.

The network was constructed using ‘signed’ adjacency to capture positive and negative correlations, critical for understanding the complex responses of citrus species and tissues to CLas under diverse conditions. Module-trait correlations (Figure 4; Appendix A) revealed significant associations between modules and key variables: *Citrus species*, *Tissue* type, and *DPI*. In contrast, the infection *Treatment* per se did not show strong or consistent correlations across modules. This may be due to the dominant influence of genotype and tissue identity, which can mask direct treatment effects in global expression patterns, especially when multiple citrus varieties and organs are combined in the analysis. Functional annotation of selected modules provided insights into potential biological roles in HLB responses.

### 2.6. Gene Significance and Module Membership Results

For the *Citrus species* trait, the co-expression analysis identified 2535 genes with a moderate to strong correlation, including 120 genes with a strong correlation. Positively correlated genes were mainly distributed in the *skyblue3* and *turquoise* modules (totaling 1622 genes), while negatively correlated genes were predominantly associated with the *darkgreen*, *salmon*, and *thistle2* modules (913 genes). Among the highly correlated genes, 18.33% (22 genes) were functionally annotated, and overall, 17.91% (454 genes) of the moderately to strongly correlated genes had functional annotations (Appendix A). This gene set likely represents key molecular players involved in species-specific responses to HLB, offering a foundation for further investigation into resilience mechanisms among citrus varieties.

The *skyblue3* module, which comprises 102 genes, displayed a strong positive correlation with the Citrus species trait (r = 0.79). A STRING analysis using *C. sinensis* as a reference and a confidence score of 0.4 revealed a functional cluster, centered on proteins containing PIR2-like helical and IPT domains. This cluster included genes such as *A0A067EIV1*, *A0A067G1B0*, *A0A067G8L6*, and *A0A067GP03* (see Appendix A). ClueGO also identified the functionally enriched gene *CISIN_1g005057mg* as a zinc ion-binding protein. Additionally, the *turquoise* module—the largest containing 5539 genes—showed a moderately positive correlation (r = 0.54). Functional enrichment analyses using STRING and ClueGO revealed their involvement in a variety of biological processes, such as metabolism, transport, regulatory functions, and signal transduction. Notable functional categories included carbohydrate and lipid metabolism, monoatomic ion transport, nucleobase-containing compound metabolism, organelle organization, protein metabolism, intracellular transport, translation, ribonucleoprotein complex biogenesis, macromolecule biosynthesis regulation, vesicle-mediated transport, and nitrogen compound transport. These processes encompassed many associated genes (see Figure 5 and Appendix A).

The three negatively correlated modules, *darkgreen* (326 genes, r = −0.59), *salmon* (596 genes, r = −0.69), and *thistle2* (567 genes, r = −0.91), were found to be enriched in processes related to genome regulation, protein modification, and mitochondrial function (see Appendix A). The *darkgreen* module was enriched in transposase activity, zinc finger domains, and nuclear pore components (e.g., GO:0031080), suggesting transcriptional and chromatin-associated functions. The *salmon* module showed an overrepresentation of N-terminal acetylation (GO:0006479) and acetyltransferase complexes, such as NatC, indicating its involvement in protein modification and co-translational regulation. In contrast, the *thistle2* module showed strong enrichment for mitochondrial import machinery, particularly the TIM23 complex (GO:0005744), as well as terms related to protein targeting and organelle localization. Together, these modules reflect distinct biological programs which may be repressed in association with *Citrus species* variation under HLB conditions.

*Tissue* trait analysis identified 2555 genes with a moderate to strong correlation, including 56 with a strong correlation. Positively correlated genes were mainly located in the *darkgrey* and *plum1* modules (totaling 496 genes), while negatively correlated genes were predominantly associated with the *lightgreen*, *yellow*, and *steelblue* modules (2059 genes). Functional annotation was achieved for 17.87% (10 genes) of the strongly correlated set and for 19.65% (502 genes) of the entire group of correlated genes (Appendix A). This gene set likely reflects tissue-specific transcriptional programs during HLB infection, potentially involving defense-related responses in leaves, metabolic reprograming in roots, or specialized signaling pathways depending on tissue identity.

The *plum1* module, consisting of 88 genes, showed a strong positive correlation with the Tissue trait (r = 0.66). This module revealed a functionally enriched STRING cluster associated with the C-terminally encoded peptide and peptide hormone binding categories. ClueGO identified the following enriched functional groups: the cytoplasmic dynein complex, transcriptional regulation via zinc ion binding, and microtubule-related processes. The genes *LOC102624055*, *LOC102625646*, and *LOC102626792* are also related to these processes (see Figure 6). The *plum1* module may participate in tissue-specific regulatory mechanisms that are possibly related to peptide signaling and intracellular transport pathways that are activated in response to HLB infection. The *darkgrey* module, comprising 318 genes, showed a moderate positive correlation with the Tissue trait (r = 0.51). STRING analysis revealed several functionally enriched clusters, including TRM32-like proteins, mostly uncharacterized proteins, deadenylation-independent decapping of nuclear-transcribed mRNA, and identical protein binding. Additional categories included protein homodimerization activity and mRNA decay pathways. Among the genes contributing to these clusters are *A0A067EQN9*, *A0A067EZH3*, and *A0A067GIZ8*. According to ClueGO, *LOC102611288* (also annotated as CysP) was associated with lysosome and cysteine-type endopeptidase activity, suggesting possible roles in protein turnover and degradation processes essential for enhancing plant defenses (see Appendix A).

The three modules that were negatively correlated with the *Tissue* trait were found to be functionally enriched in distinct yet complementary biological processes. This suggests the existence of coordinated transcriptional programs across specific genotypes. These modules were labelled *lightgreen* (1461 genes, r = –0.57), *yellow* (2077 genes, r = –0.56), and *steelblue* (188 genes, r = –0.76). The *lightgreen* module showed significant enrichment in pathways related to lipid metabolism (cit00071), the biosynthesis of secondary metabolites (cit01110) and the signaling of oxylipins, notably including the metabolism of alpha-linolenic acid (cit00592) and linoleic acid-derived compounds (CL:9792). The yellow module showed significant enrichment in primary metabolic pathways, such as starch and sucrose metabolism (cit00500), galactose metabolism (cit00052), and oxidative phosphorylation (cit00190). ClueGO analysis of this module’s functional landscape also identified secondary metabolite biosynthesis (genes *LOC102606812*, *LOC102625429*, and *LOC102628384*) and amino sugar metabolism (genes *CHI1*, *LOC102617819*, and *LOC102628588*), indicating a central role in carbon allocation and energy production. Overall, this module likely reflects a metabolically active transcriptional network involved in tissue differentiation and responsiveness to physiological cues. The *steelblue* module displayed the strongest negative correlation with variety and was functionally specialized in phenylpropanoid biosynthesis (cit00940) and oxidative stress response mechanisms involving peroxidase activity (GO:0004601) in particular. This includes genes associated with hydrogen peroxide metabolism (GO:0042744), flavonoid biosynthesis (STRING cluster CL40574), and ethylene signaling (*LOC102622241*). This supports a model of transcriptional activation related to plant defense and detoxification processes. ClueGO identified key genes involved in phenylpropanoid biosynthesis (*LOC102622229*) and the ethylene-activated signaling pathway (*LOC102622241*), indicating the presence of an immune-responsive module that may be involved in the hypersensitive response (HR) and systemic acquired resistance (SAR)-like pathways in response to HLB infection or genotype-specific stress perception. Taken together, these three negatively correlated modules may represent variety-associated transcriptional signatures involving metabolic reprograming.

Regarding the *DPI* trait, the co-expression analysis revealed 4431 genes with moderate to strong correlations, including 8 genes showing strong positive correlations. The positively correlated genes were primarily found in the *floralwhite*, *orangered4*, *plum1*, and *turquoise* modules (totalling 2480 genes), whereas the negatively correlated genes (1951 in total) were predominantly associated with the *brown*, *darkgrey*, and *salmon* modules. Despite the limited number of genes showing strong positive correlation, 25% of these (2 genes) were functionally annotated. Of all the genes showing moderate to strong correlations, 19.07% (845 genes) were functionally annotated (see Appendix A).

The *floralwhite* and *orangered4* modules, comprising 1092 and 74 genes, respectively, showed moderate positive correlations with *DPI* (r = 0.56 and r = 0.54, respectively). STRING annotation for the *floralwhite* module revealed enrichment in intracellular membrane-bounded organelles (GO:0043231), membrane organization (GO:0071840), and RNA processing (GO:0016070) (Appendix A). Notably, several genes were associated with SNARE complex assembly and vesicle docking, including *LOC102621665* and *LOC102627357*. ClueGO analysis further supported these findings, revealing significant enrichment in biological processes such as the regulation of RNA metabolic process (e.g., *LOC102577939*; *LOC102622681*), monoatomic ion and organic anion transmembrane transport (e.g., *LOC102608041*, *LOC102608591*, and *LOC102628385*), and auxin-activated signaling pathway (e.g., *LOC102612823*; *LOC102627489*) (Figure 7). The *orangered4* module showed enrichment for protein clusters containing transposase-like domains, zinc fingers, and FAR1 DNA-binding motifs (CL:39473), with several genes (e.g., *A0A067E9B0, A0A067EJY8,* and *A0A067EKA9*) recurrent across top-enriched clusters, suggesting roles in genome regulation or chromatin remodeling. Additional enrichment in sugar/inositol transporters (CL:46591, InterPro IPR003663) and aminoacyl-tRNA synthetase activity (CL:1570) suggest involvement in nutrient transport and translation. ClueGO results supported these findings, highlighting *LOC102611476* (asparagine-tRNA ligase activity) and *LOC102612876* (sugar transmembrane transporter activity), reinforcing the dual metabolic and transport-related profile of this module during infection. The *turquoise* (r = 0.55) and *plum1* (r = 0.41) modules also showed moderate positive correlations with *DPI* and have been functionally described in detail in previous sections.

The *brown* module, which comprises 2306 genes, exhibited a moderate negative correlation with DPI (r = −0.43). This indicates that the expression of its genes tends to decrease as the infection progresses. KEGG pathway analysis identified Phenylpropanoid biosynthesis (cit00940) as the most significantly enriched pathway (FDR = 1.37 × 10^−10^). This pathway includes key enzymes, such as peroxidases and other oxidoreductases (e.g., *A0A067EBI5, A0A067FS47*, and *A0A067GVX6*). Furthermore, GO and STRING annotations revealed enrichment in genes associated with lignin biosynthesis (GO:0009808), hydrogen peroxide metabolism (UniProt KW-0376), and proteins with zinc finger domains (IPR006564) involved in transposase activity.

The *turquoise* (r = 0.55) and *plum1* (r = 0.41) modules showed positive correlations with DPI, while the *darkgrey* (r = −0.57) and *salmon* (r = −0.41) modules showed negative correlations. All four modules have already been characterized in the context of *Tissue* and *Citrus species* traits.

This gene set is likely to be involved in temporal responses to HLB progression, given that *DPI* reflects time post-infection. Together with the results obtained for *Tissue* and *Citrus species*, this analysis provides a comprehensive framework for understanding gene expression changes during infection dynamics.

### 2.7. Differential Gene Expression Analysis Across Selected Citrus Bioprojects

After identifying the modules most closely related to the selected variables—*Citrus species* (*skyblue3* and *turquoise*), *DPI* (*floralwhite*, *orangered4*, *plum1*, and *turquoise*), and *Tissue* type (*darkgrey* and *plum1*)—we then examined differential gene expression across various conditions, treatments, citrus species, and tissues by selecting six relevant Bioprojects: (1) 203307, which included mature and immature fruit and leaf samples from orange plants; (2) 417324, a study on infected versus uninfected orange leaves sampled at 56, 126, 182, and 322 dpi; (3) 557834, which analyzed leaf and spine samples from grapefruit plants with (W) and without thorn (TL), including mutants (T); (4) 629966, an analysis of orange feeder root samples taken at 0, 3, and 9 dpi, with Hoagland solution supplementation post-infection; (5) 739186, which analyzed symptomatic (S) versus asymptomatic (A) infected leaf samples of mandarin across four seasons (fall, spring, summer, and winter); and (6) 755969, a comparison of infected leaf samples between finger lime and orange (extra data could be found in Appendix A). Reads from each dataset were processed independently through low-quality read removal, mapping against the *C. sinensis* DHSO v3.0 genome, read counting, and differential gene expression analysis with DESeq2 as detailed in the Materials and Methods. Differentially expressed genes (DEGs) were filtered based on the modules of interest (*floralwhite*, *orangered4*, *plum1*, *turquoise*, *skyblue3*, and *darkgrey*) and then categorized into up and downregulated genes. An UpSet plot (Figure 8) was generated to highlight shared DEGs across different conditions, with panel A illustrating consistently upregulated genes and panel B showing consistently downregulated genes across multiple experimental conditions (Appendix A).

Regarding DEGs associated with the *Citrus* species trait, a subset of 49 genes was consistently upregulated in comparisons between different species, particularly in Bioproject 755969. Notably, 3 of these genes were shared with tissue, and 25 were shared with the DPI. Functional enrichment analysis—supported by network-based inference—revealed a significant over-representation of biological processes related to flavin metabolism, including the riboflavin biosynthetic process (GO:0009231) and the FAD metabolic process (GO:0046443). Additional enriched terms included oxidoreductase activity acting on the CH-OH group of donors (GO:0016614), FMN binding (GO:0010181), and (S)-2-hydroxy-acid oxidase activity (GO:0003973), suggesting species-specific metabolic adjustments in redox homeostasis and small molecule processing. These functions were inferred through the integration of DEGs with their proximal interactors in the STRING network.

A total of 110 downregulated genes were identified in *Citrus species* using Bioproject 755969. Of these, 19 are also associated with variable *Tissue*, indicating an overlap between tissue-specific expression patterns. These genes are enriched in metabolic processes related to lipid metabolism and RNA regulation. Key biological processes include the catabolic process of coenzyme A (GO:0015938), which involves four genes that are essential for breaking down CoA derivatives, and the biosynthetic process of fatty acids (GO:0006633), which involves seven genes that contribute to membrane synthesis and signaling. Organophosphate catabolism (GO:0046434), with five genes, suggests a role in phosphate recycling. The modules also include three genes involved in ta-siRNA processing (GO:0010267), pointing to regulatory roles in gene silencing. At the molecular function level, five genes exhibit acyl-[acyl-carrier-protein] desaturase activity (GO:0045300), influencing membrane fluidity and stress responses. Four genes are associated with acetyl-CoA hydrolase activity (GO:0003986) and CoA pyrophosphatase activity (GO:0010945), reflecting control over key metabolic intermediates.

Examining the *DPI* trait, we first explored whether early signs of transcriptional reprograming could be detected prior to the advanced symptomatic stage by analyzing samples from 56 DPI in the susceptible Valencia orange genotype. Functional inference based on network proximity in STRING suggested the early activation of lipid-related processes, including the CDP-diacylglycerol biosynthetic process (GO:0016024), phosphatidylglycerol biosynthesis (GO:0006655), and various acyltransferase-related molecular functions. These pathways converge on membrane lipid remodeling and chloroplast-associated metabolism, potentially reflecting an early attempt to maintain cellular homeostasis under stress. In contrast, at 322 DPI in Bioproject 417324, we identified 16 upregulated DEGs with significant enrichment in trehalose metabolism (GO:0005992) and cellular carbohydrate biosynthesis (GO:0034637). Six genes correspond to isoforms of *Cs_ont_5g041790.x*, while others—inferred by association—include *Cs_ont_9g023730.1/2* and *Cs_ont_1g000570.2*; *Cs_ont_3g020540.4* was directly identified as a DEG. All encode trehalose phosphatases, enzymes that catalyze the final step of trehalose biosynthesis by converting trehalose-6-phosphate into trehalose, a disaccharide associated with stress protection. Accordingly, molecular function analysis highlighted significant enrichment in trehalose-phosphatase activity (GO:0004805), reinforcing the central role of this pathway. Additionally, subcellular localization terms such as plasmodesma (GO:0009506), endoplasmic reticulum (GO:0005783), cell periphery (GO:0071944), and plastid (GO:0009536) were over-represented, suggesting coordinated modulation of intercellular communication, protein secretion, and plastidial metabolism during late infection stages. Collectively, these results indicate a transcriptional program oriented toward metabolic reconfiguration and cellular protection in response to prolonged pathogen exposure.

In the *DPI* downregulation analysis, four genes from Bioproject 417324 at 56 DPI were functionally linked to secondary metabolism pathways, particularly those related to phenylpropanoid biosynthesis (map00940), stilbenoid and flavonoid biosynthesis (map00945, map00941), and cytochrome P450-associated oxidative processes (Reactome MAP-211897 and MAP-211859). In the same Bioproject at 322 DPI, 52 genes were significantly repressed, reflecting late-stage HLB infection in orange leaves. Of these, 27 overlapped exclusively with Bioproject 755969 (finger lime vs. orange), and 14 overlapped with both 755969 and 739186 (spring season mandarin). Functional enrichment analysis revealed strong repression of key metabolic and physiological processes, particularly those related to membrane lipid metabolism and chloroplast function. Five isoforms of *Cs_ont_3g001000* were central to the suppression of membrane lipid metabolic processes (GO:1905038), sterol transport (GO:0015918), and lipid biosynthesis regulation (GO:0046890). Additionally, 7 genes, including *Cs_ont_2g009080.1* and *Cs_ont_5g026340.1*, were associated with photosynthesis (GO:0015979), while 24 genes—nearly half of the downregulated set—were annotated to the chloroplast (GO:0009507), underscoring HLB’s detrimental effects on photosynthetic capacity. Enrichment of molecular functions such as sterol binding (GO:0032934) and lipid transporter activity (GO:0005319) further supports disrupted membrane dynamics and signaling. Collectively, this signature reveals the collapse of chloroplast-associated metabolism, sterol transport, and biosynthesis during advanced infection.

Studying DEGs in the *Tissue* trait, we found 19 upregulated genes in Bioproject 557834 (grapefruit leaf and spine samples), in the T vs. W comparison, of which 14 were also shared with the TL vs. W interaction, and 1 gene overlapped with Bioproject 755969. Enrichment was biased toward photosynthesis and chloroplast processes—Photosynthesis (GO:0015979), Photosynthesis light reaction (GO:0019684), Photosynthetic electron transport in PSII (GO:0009772), Photosystem II (GO:0009523), and Chloroplast thylakoid membrane (GO:0009535)—and translational regulation (rRNA pseudouridine synthesis, GO:0000455; aminoacyl-tRNA editing, GO:0002161), suggesting fine-tuning of protein synthesis and photosynthetic machinery in T and TL relative to W.

Bioproject 755969 revealed that 72 genes that were significantly downregulated were associated with the modules related to the trait *Tissue*. Of these, 19 were also present in the *Citrus species* trait, and 15 of these were linked to *DPI*. Moreover, 12 of the 72 genes were also present in Bioproject 417324 at 322 DPI, and 3 were in a sample collected in spring from plants infected with HLB (Bioproject PRJNA739186). Lipid-related pathways dominated enrichment: CDP-diacylglycerol biosynthesis (GO:0016024; 13 genes), phospholipid metabolism (GO:0006644; 20 genes), glycerophospholipid metabolism (GO:0006650; 18 genes), and phospholipid biosynthesis (GO:0008654; 15 genes). Molecular functions such as 1-acylglycerol-3-phosphate O-acyltransferase (GO:0003841; 10 genes) and O-acyltransferase (GO:0008374; 14 genes) were highly significant. KEGG highlighted glycerophospholipid metabolism (map00564; 21 genes) and glycerolipid metabolism (map00561; 15 genes), while Reactome confirmed suppression of phosphatidic acid synthesis (MAP-1483166; 15 genes) and glycerophospholipid biosynthesis (MAP-1483206; 16 genes). Overall, this signature reflects profound reprograming of lipid metabolism in finger lime versus orange, partially conserved across infection stages and genotypes.

## 3. Discussion

Numerous studies have investigated the response of citrus to HLB disease [12]. Over the past 15 years, the increasing use of large-scale plant sequencing technologies has enabled significant advances in transcriptome analysis, particularly through RNA sequencing [51,52]. Among these, the study by Martinelli et al. [53] marked a pivotal transition from microarray to RNA-seq technologies, providing deeper insights into citrus–pathogen interactions. The increased public availability of transcriptomic datasets has also facilitated co-expression analysis, which critically depends on the number of available samples, data quality, and experimental design—factors that are essential for maximizing statistical power and minimizing the likelihood of false positives [54]. While in silico studies exploring the interaction between CLas and Citrus have been reported [48,55], this work provides the first comprehensive analysis integrating multiple citrus species, diverse tissues, and different infection stages. Previous studies relied on narrower datasets, such as metabolic pathway analysis using 13 samples from three Bioprojects [55], or co-expression networks using 66 samples from six Bioprojects, primarily using differential expression analysis to assess gene function [48].

In this study, data from 17 bioprojects (293 samples) were integrated to construct a global co-expression network of citrus plants affected by HLB. This large-scale approach allowed the identification of 47 biologically meaningful modules associated with key experimental factors, including citrus species (e.g., sweet orange and mandarin), tissue type (e.g., leaves, fruits, and roots), and DPI, representing different stages of HLB progression. By leveraging a significantly larger and more diverse dataset, this study provides a deep understanding of citrus–pathogen interactions and highlights potential gene regulatory hubs and pathways relevant to HLB disease progression.

The integrative co-expression and DEG analyses for the *Citrus species* trait identified key modules underpinning differential tolerance to HLB. Notably, modules enriched in PIR2-like domains linked to positive modulation of abscisic acid (ABA) signaling [35,56] and isopentenyl transferases (IPTs) suggest the involvement of complex hormonal regulation during infection [57]. While bacterial IPTs elevate auxin to promote pathogen colonization, host auxin biosynthesis genes are typically repressed under CLas infection; however, tolerant genotypes appear to maintain higher levels of growth-related hormones, such as auxins and cytokinins, thereby supporting vascular regeneration and mitigating phloem [58]. HLB progression is further associated with micronutrient imbalances, evidenced by zinc-associated genes in the *skyblue3* module. Systemic micronutrient depletion has been reported in infected trees, notably zinc in leaves and boron in roots, due to impaired phloem transport [26,59,60], which further exacerbates nutrient deficiencies [61,62]. Tolerant species may attenuate these deficiencies through sustained growth and phloem regeneration [58].

HLB also induces substantial metabolic reprogramming. Abnormal starch accumulation and altered carbohydrate metabolism disrupt the source–sink dynamic [53], while perturbations in lipid biosynthesis and β-oxidation compromise membrane integrity and signaling, enhancing stress responses [63]. Our data corroborates this, with upregulation of riboflavin biosynthesis (GO:0009231) and FAD metabolic process genes (GO:0046443), suggesting increased production of flavin cofactors to support redox homeostasis, a role substantiated in other plant–pathogen systems [46]. Conversely, downregulated genes were enriched in fatty acid biosynthesis (GO:0006633) and CoA catabolism (GO:0015938), implying reduced capacity for membrane synthesis and metabolic flexibility. These transcriptional patterns, observed in the tolerant finger lime, are consistent with a reprograming strategy that favors the synthesis of specialized defense lipids over general lipid metabolism [14,48,64,65]. Moreover, finger lime increased the expression of antioxidant enzymes (e.g., peroxidases and glutathione-S-transferases), which help mitigate ROS-induced damage [14] and maintain energy homeostasis, in contrast to the metabolic exhaustion observed in susceptible cultivars [14]. Epigenetic and post-transcriptional regulation also contribute to HLB responses; tolerant Mexican lime exhibits extensive miRNA reprograming, which modulates defense and hormone signaling [66], while specific miRNAs (e.g., *miR171b*) have been shown to positively regulate HLB resistance [67]. We further identified downregulation of ta-siRNA processing genes (GO:0010267), indicating compromised gene silencing under HLB stress. Disruption of small RNA pathways may impair the fine-tuning of stress responses, as shown in other studies [68]. Finally, mitochondrial adjustments appear pivotal: tolerant cultivars maintain respiratory activity, whereas susceptible ones exhibit mitochondrial dysfunction and energy deficits. Infected tissues accumulate ATP and ROS in correlation with symptom severity, while tolerant genotypes enhance their antioxidant defenses [69]. Consistently, the *thistle2* module, negatively correlated with *Citrus species*, was enriched for mitochondrial import machinery (TIM23 complex), although no associated DEGs were identified in the finger lime vs. sweet orange comparison (Bioproject 755969).

The transcriptional landscape associated with the *DPI* trait reflects a coordinated temporal response to HLB progression across diverse citrus varieties. Co-expression analysis revealed that key modules were enriched in processes related to vesicle trafficking, membrane organization, and chromatin remodeling, supporting the existence of an orchestrated regulatory program during infection [55,70]. To further explore the temporal dynamics of this response, differential expression analysis was conducted at both 56 and 322 DPI using samples from the highly HLB-susceptible Valencia orange [71], offering insights into both the initial transcriptional adjustments and the extensive reprograming that characterize advanced stages of the disease.

At 56 DPI, functional inference based on module–trait relationships and guilt-by-association networks suggests a preparatory state characterized by the activation of lipid biosynthetic processes—particularly those involving chloroplast membrane precursors and acyltransferase activity [20,48,64]. These early transcriptional changes likely represent compensatory mechanisms aimed at preserving organelle integrity and sustaining photosynthetic capacity [72,73]. However, by 322 DPI, this response appears overwhelmed: direct DEG analysis revealed repression of genes related to photosynthesis and chloroplast function, along with downregulation of sterol transport and membrane lipid metabolism genes, indicating structural deterioration and energetic collapse [55,65,74]. In contrast, guilt-by-association analysis suggested the activation of trehalose biosynthesis at this advanced stage, which may reflect a shift toward stress mitigation and the mobilization of carbohydrate reserves. However, this interpretation lacks direct support from existing physiological or metabolomic studies in HLB-infected citrus. Instead, reported stress responses include the accumulation of starch [74], free proline [13,38], and soluble sugars [38] which act as established osmoregulatory metabolites in infected tissues. Therefore, while the trehalose pathway may represent a network-inferred association with general stress responses, its functional role in HLB progression remains to be experimentally confirmed. Together, the co-expression and DEG results support a sequential pattern of response: from early buffering and maintenance to late-stage damage control and metabolic reallocation, highlighting the temporal vulnerability of susceptible citrus genotypes such as ‘Valencia’ orange [19,35,73,74].

Unlike the time-dependent progression of HLB, tissue-specific responses reveal a mosaic of localized transcriptional strategies shaped by organ identity and function. Our analysis of the Tissue trait exposed distinct expression patterns across citrus organs and genotypes [48,75]. Previous studies have shown that HLB elicits contrasting gene expression programs depending on the tissue type, developmental stage, and experimental conditions, with young and mature leaves, roots, and fruits exhibiting distinct transcriptional and hormonal profiles [21,35,53,55,70,75,76,77]. Such divergence likely reflects the pathogen’s ability to differentially disrupt source–sink relationships and modulate tissue-specific defense signaling pathways, including salicylic acid-, jasmonic acid-, and abscisic acid-mediated responses [20,26,72,77]. These context-dependent transcriptional programs likely contribute to tissue-specific susceptibility or resilience and help explain the low overlap in differentially expressed genes reported across independent transcriptomic studies [20,21,35,41,53,64,78,79,80,81]. Our results complement these observations by showing that modules positively correlated with *Tissue* trait identity were enriched in intracellular transport, transcriptional regulation, and peptide hormone signaling—suggesting tissue-specific regulatory programs potentially involved in long-distance signaling or cell-to-cell communication. In contrast, negatively correlated modules showed coordinated repression of lipid metabolism, photosynthesis, and phenylpropanoid biosynthesis. These patterns were supported by DEG analysis, which revealed downregulation of lipid biosynthetic and photosynthetic genes in infected finger lime and orange, while several upregulated genes in tolerant grapefruit tissues were associated with functions potentially linked to chloroplast maintenance and protein translation. The spatial dimension of HLB responses emphasizes that resilience is not uniformly distributed across the plant. Instead, it emerges from organ-specific programs that balance protective investment with metabolic cost. By integrating DEG and co-expression signals, our study points to new opportunities for designing tissue-targeted interventions that reinforce structural and physiological defense in critical organs.

These findings highlight the importance of transcriptional reprograming, metabolic adaptation, and protein turnover in shaping the plant’s response to HLB infection. The integration of these mechanisms underscores the complexity of host–pathogen interactions and offers valuable insights into potential targets for early diagnostic and management strategies [70,78]. Furthermore, the significant proportion of unannotated genes within the modules most strongly associated with citrus species, tissue identity, and infection stage underscores the need for further research to explore their possible roles in defense. Together, these findings emphasize the central importance of these module–trait associations in orchestrating stress responses and pathogen resistance, while providing a promising framework for future studies aimed at unraveling their uncharacterized genetic components and their potential role in HLB tolerance mechanisms.

## 4. Materials and Methods

### 4.1. Data Acquisition

We selected 17 bioprojects related to HLB treatment and various *Citrus* spp., along with other treatment conditions. A total of 293 RNA-seq samples were obtained from the Sequence Read Archive (SRA) (NCBI), spanning five Citrus species or hybrids: grapefruit (*Citrus* × *paradisi*), lemon (*Citrus* × *limon*), lime (*Citrus* × *aurantiifolia*), mandarin (*Citrus reticulata*), and sweet orange (*Citrus* × *sinensis*). These samples cover nine tissue types, including buds, calyx abscission zone, fruit, leaf, midribs, thorn, feeder roots, roots, and protoplasts derived from embryogenic calli. Detailed information on the SRA studies and associated metadata can be found in Appendix A.

### 4.2. Quality Control and Mapping of Data

Sequence quality for each sample was assessed using FastQC [82]. Adaptors and low-quality reads were removed with Trimmomatic v0.39 [83]. The samples were aligned to both *Citrus sinensis* DHSO v3.0 [84] and *Citrus clementina* v1.0 [85] reference genomes, and mapping efficiency was evaluated based on uniquely mapped reads for STAR v2.7.10b [49] and the ‘aligned concordantly exactly 1 time’ metric for HISAT2 v2.2.1 [86] aligners with default parameters. This multi-alignment approach was used to compare alignment percentages between the clementine and sweet orange genomes, and the genome with the highest alignment efficiency was selected for further analysis. Samples with less than 60% alignment were excluded. Gene expression levels were quantified using featureCounts v2.0.3 [87], assigning counts to ‘transcript_id’ instead of ‘gene_id’, as the goal of this study was to focus on transcript-level gene expression.

### 4.3. Data Normalization and Outlier Detection

Count data were normalized using VST from the DESeq2 R package v1.30.1 [88], to reduce the influence of highly expressed genes and stabilize variance across expression levels. To correct batch effects associated with the *Bioproject* variable—which reflects differences in sequencing platform, location, and other technical factors—we applied the ComBat-seq method [89] from the sva R package v3.54.0 in the Bioconductor project [90]. This method models batch effects in count data using a negative binomial approach and preserves the integer nature of the adjusted data, allowing its direct use in downstream analyses.

Outlier detection was subsequently performed using hierarchical clustering with Pearson correlation as the similarity metric. Sample dissimilarity was defined as 1 − correlation, and clustering was conducted using the average linkage method. A dendrogram was generated to visualize sample relationships, and a height cutoff of 0.5 was applied to identify and exclude outlier samples. This filtering step was essential to reduce technical variability and retain high-quality data for subsequent analyses, including differential expression and co-expression network construction. A schematic overview of the full analysis workflow is presented in Appendix A.

### 4.4. Weighted Gene Co-Expression Network Analysis

A weighted gene co-expression network analysis (WGCNA) was performed to identify clusters of highly correlated genes and potential regulatory candidates or biomarkers [44]. The analysis was conducted using the WGCNA R package (v1.7.1). Pearson correlation was initially used as a similarity measure to calculate adjacency, which was then transformed into a signed topological overlap matrix (TOM) to minimize noise and spurious associations. The network construction parameters, such as soft-thresholding power (β), merge cut height, and minimum module size, were optimized to fit a scale-free topology. The soft-thresholding power was selected using the function pickSoftThreshold, which analyzes network topology to choose an appropriate power. Based on the approximate scale-free topology criteria, we selected a power suitable for a high scale-free topology fit (R^2^ ≥ 0.90) with an acceptable level of network connectivity. For module detection, the deep split parameter was set to 2, the merge cut height was set to 0.25, and the minimum module size was set to 30 genes. This process allowed for the identification of distinct modules, which were further analyzed for biological relevance.

### 4.5. Gene Significance (GS) and Module Membership (MM) Evaluation

To evaluate the association of each gene with traits of interest (citrus species, tissue, days post-infection (DPIs), and treatment), Gene Significance (GS) was defined as the correlation between the gene expression and the trait in question. Similarly, Module Membership (MM) was calculated as the correlation between the module eigengene and each gene’s expression profile. These values allow us to assess the importance of genes within their respective modules and their relationship with the traits under study.

For module–trait relationships, we considered correlations with an absolute value of r ≥ 0.4 and *p*-value < 0.05 to be biologically meaningful. This threshold was chosen to highlight moderate to strong associations that may underline relevant biological processes, while avoiding spurious correlations.

### 4.6. Functional Annotation and Network Visualization

The protein sequences of *Citrus sinensis* DHSO v3.0 were initially unannotated. To address this, we used the Uniprot protein database specific to *Citrus sinensis* (taxonomy_id: 2711, https://www.uniprot.org/uniprotkb?query=%28taxonomy_id%3A2711%29, accessed on 1 September 2024). Protein alignments were conducted using the blastp command from BLAST+ v2.16.0 to identify the best matches, selecting candidates based on the lowest e-values, with a cut-off set at 0.001. This approach enabled us to integrate high-quality, curated data from Uniprot into our dataset, effectively linking Uniprot identifiers to our protein sequences, as no direct mapping to UniprotID or entrezgeneID was previously available. The annotated and merged dataset was subsequently used for downstream analysis. For functional annotation and exploration of protein–protein interaction networks, we utilized Cytoscape v3.10.2 [91], along with stringApp v2.1.1 [92] and ClueGO v2.5.10 [93], integrated with CluePedia v1.5.10 for enhanced pathway insights. These analyses focused on modules associated with key study variables—*DPI*, *Citrus species*, and *Tissue*—all of which showed strong correlations with gene expression patterns. Shared DEGs across different conditions in each Bioproject were identified, and their intersections were visualized using an UpSet plot generated with the UpSetR package [94], highlighting genes that are consistently differentially expressed across varying conditions. These shared DEGs were subsequently analyzed for functional interactions and annotations using the previously described protein dataset and STRING-based network tools. Additionally, when certain DEGs lacked direct functional annotations, they were linked to proximal genes using the annotated *Citrus sinensis* DHSO v3.0 proteome in STRING (taxon identifier: STRG0A34EFH, available at: https://version-12-0.string-db.org/organism/STRG0A34EFH, accessed on 14 February 2025) to infer putative functions through guilt-by-association. This was conducted solely for interpretative purposes, and these additional nodes were not incorporated into the quantitative analyses.

## 5. Conclusions

This study presents a comprehensive co-expression network analysis of HLB-affected citrus plants, providing new insights into the molecular basis of the disease. The identification of 47 biologically significant modules associated with key experimental traits—citrus species, tissue identity, and infection stage—underscores key pathways implicated in species-specific defenses, tissue-specific metabolic adaptations, and dynamic infection responses. Functional analyses highlight the critical roles of oxidative stress regulation, photosynthesis, and transcriptional reprograming in HLB progression and tolerance. These findings provide a valuable resource for citrus breeders, enabling them to prioritize genes and pathways associated with HLB tolerance in breeding programs. This foundational research paves the way for developing innovative strategies for HLB management, such as employing gene editing to enhance plant defenses. To advance our understanding of HLB pathogenesis, further experimental validation of the functional significance of these co-expression modules is essential.

## Figures and Tables

**Figure 1 plants-14-01792-f001:**
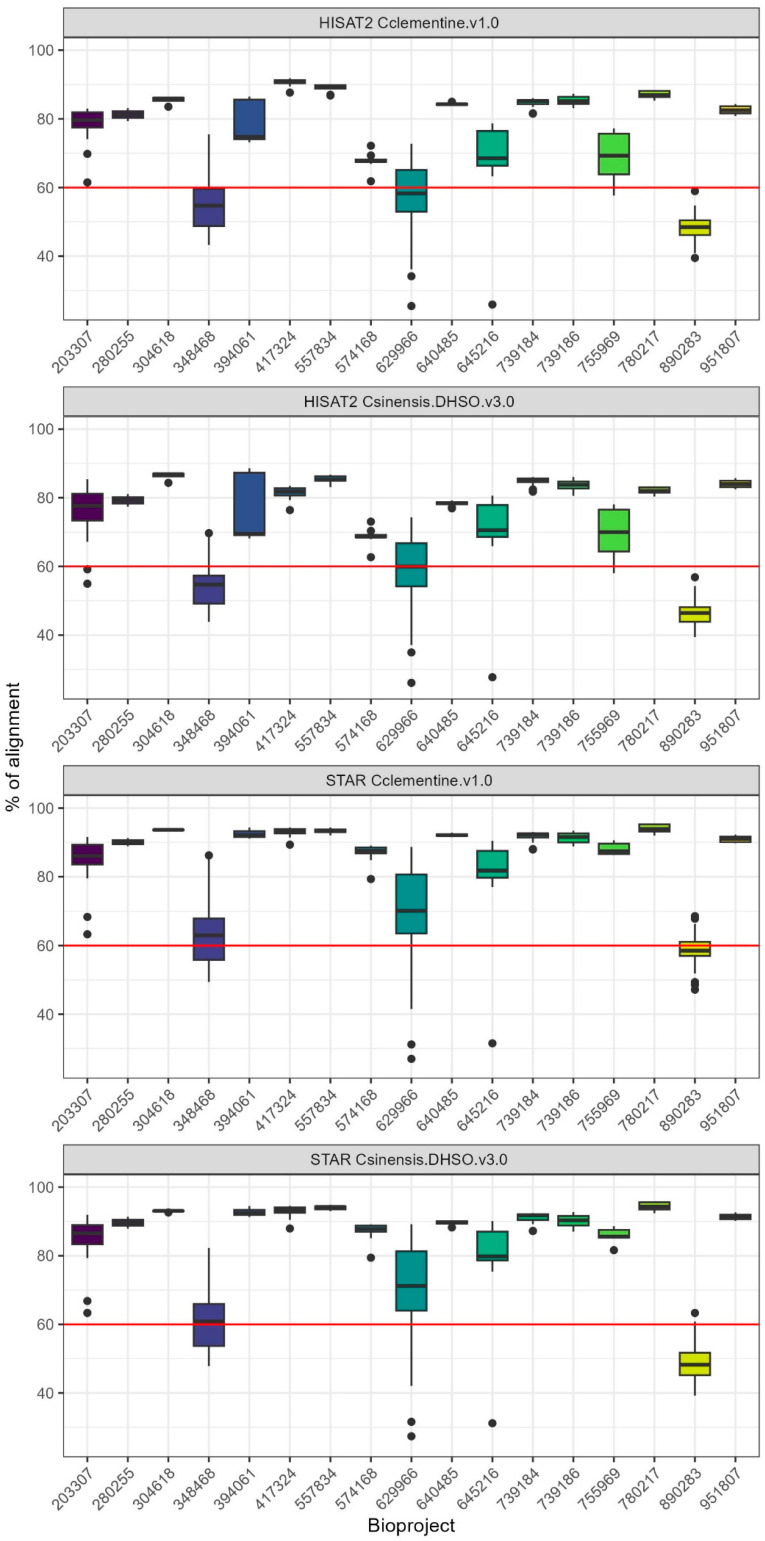
Comparison of alignment efficiency using HISAT2 and STAR aligners with *C. clementina* v1 and *C. sinensis* DHSO v3 reference genomes. Boxplots show the percentage of aligned reads across different Bioprojects. The red line marks the 60% alignment threshold, below which samples were excluded from downstream analyses. HISAT2 and STAR aligners were tested, and samples with less than 60% alignment (highlighted by the red line) were discarded to ensure data quality.

**Figure 2 plants-14-01792-f002:**
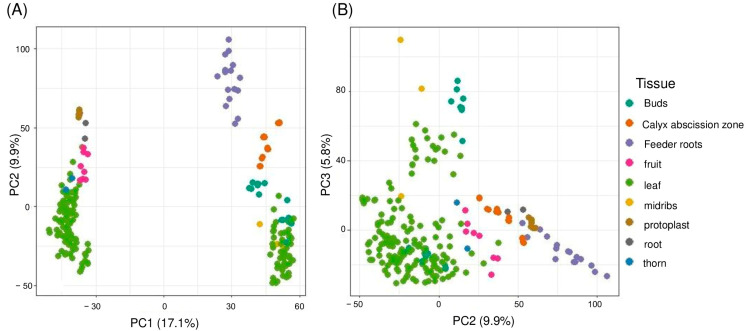
PCA of 230 RNA-seq samples after ComBat-seq batch correction. (**A**) PC1 vs. PC2 shows tissue-specific clustering. (**B**) PC2 vs. PC3 further separates tissues, confirming effective batch correction.

**Figure 3 plants-14-01792-f003:**
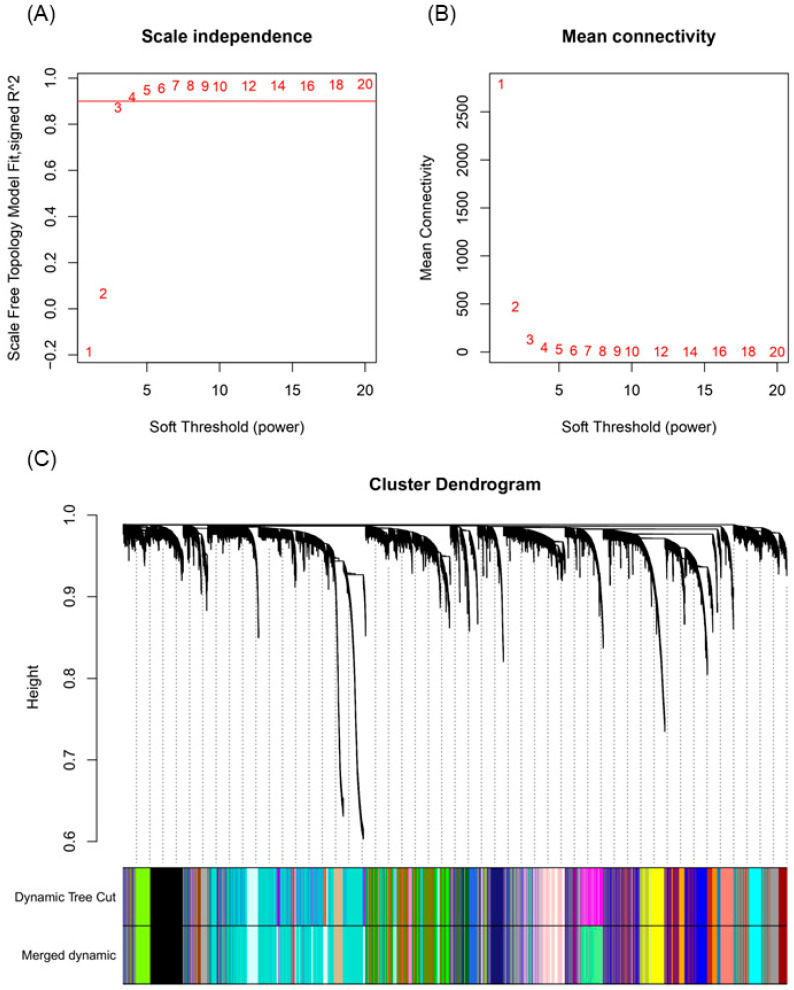
Selection of soft-thresholding power and module merging in the co-expression network analysis. (**A**) Scale-free topology fit index (R^2^) across soft-thresholding powers. The red numbers indicate the powers tested; the red line marks the R² = 0.9 reference threshold. (**B**) Mean connectivity for each power. Together, these plots guided the selection of power 7 as a balance between scale-free topology and network connectivity. (**C**) Module merging process, reducing module count to 47 based on eigengene correlation.

**Figure 4 plants-14-01792-f004:**
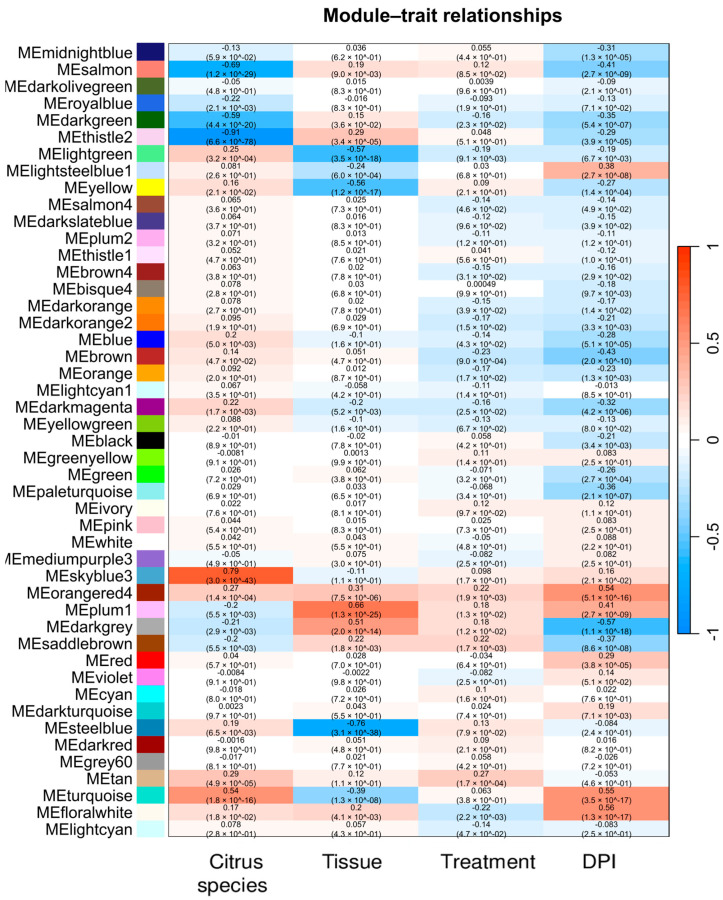
Module–trait relationships: Heatmap showing the correlation between module eigengenes and biological traits of interest, including *Citrus species*, *Tissue*, *Treatment*, and *DPI*. Each cell contains the correlation coefficient (r) and associated *p*-value, with color intensity reflecting the strength and direction of the correlation (red for positive and blue for negative correlations).

**Figure 5 plants-14-01792-f005:**
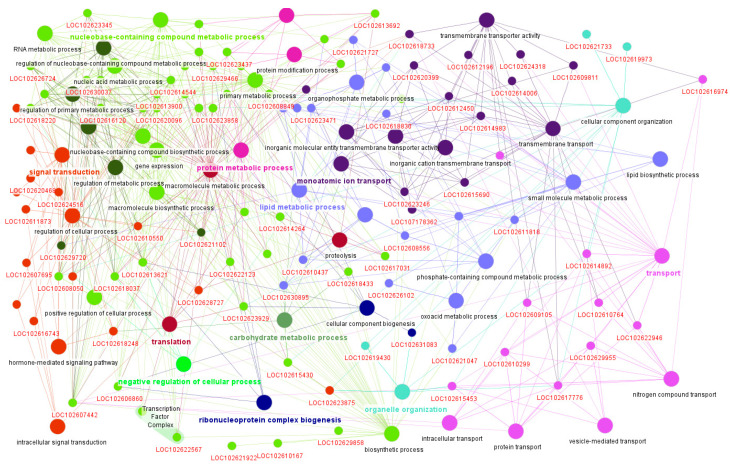
Functional enrichment analysis of key genes in the *turquoise* module using ClueGO. The analysis revealed significant enrichment in pathways related to metabolic reprograming, signal transduction, and active transport, suggesting their involvement in adaptive or response mechanisms. These functional categories may underline species-specific responses to HLB in citrus. Genes in red are associated with these enriched terms.

**Figure 6 plants-14-01792-f006:**
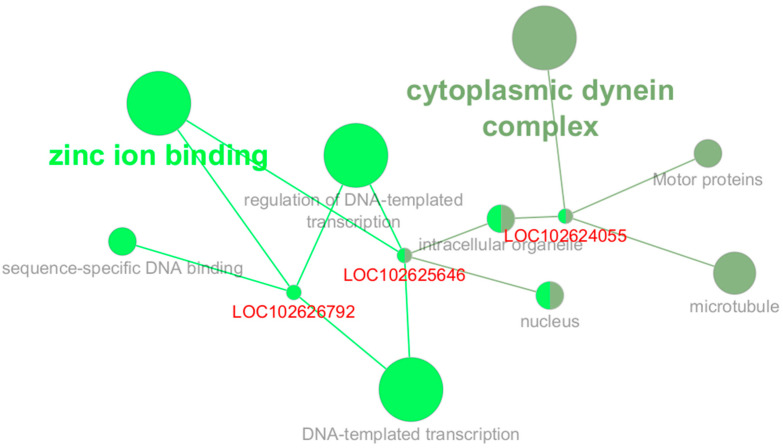
ClueGO enrichment network for the *plum1* module showing functional groups related to zinc ion binding, dynein complex, and transcription. Genes in red are associated with these enriched terms.

**Figure 7 plants-14-01792-f007:**
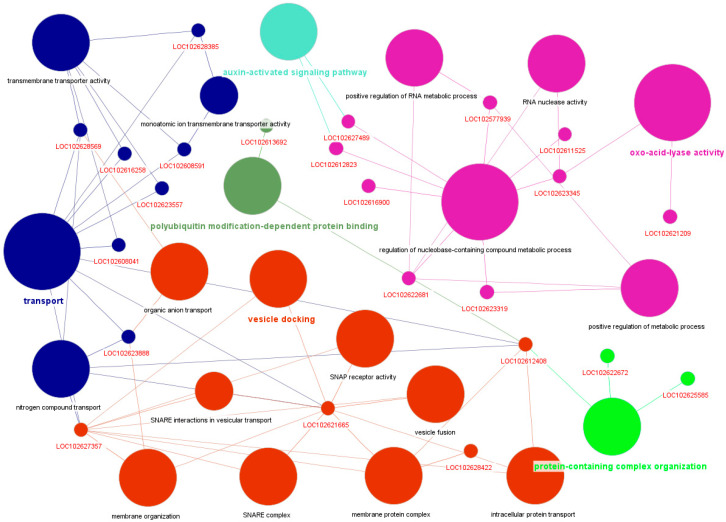
Functional enrichment of the *floralwhite* module linked to *DPI* in HLB-infected *Citrus* spp., highlighting its involvement in transport, oxoacid lyase activity, organization of protein-containing complexes, vesicle docking, and the auxin-activated signaling pathway.

**Figure 8 plants-14-01792-f008:**
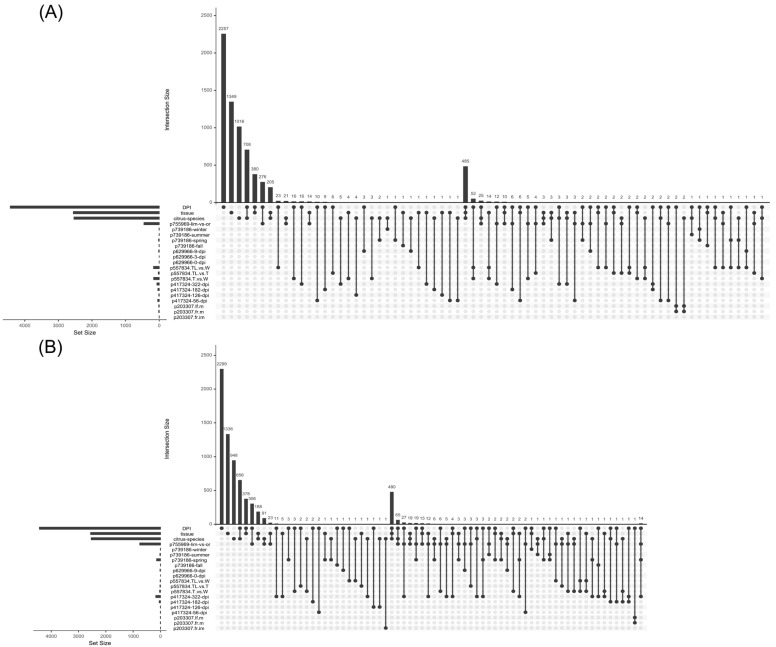
Upset diagram of differentially expressed genes: (**A**) upregulated genes; (**B**) downregulated genes. Consistent gene expression patterns are shown across different traits (*DPI*, *Citrus species*, and *Tissue*), revealing possible common pathways. Comparisons are labeled as follows: p203307_fr_im, p203307_fr_m, and p203307_lf_m represent orange fruit and leaf samples mature and immature; p417324-56-dpi, p417324-126-dpi, etc., indicate infected orange leaves at different days post-infection; p557834_T_vs_W, p557834_TL_vs_W, and p557834_TL_vs_T compare thornless, wild-type, and thorny grapefruit plants; p629966-0-dpi, p629966-3-dpi, and p629966-9-dpi represent orange feeder roots post-infection; p739186-fall, p739186-spring, etc., denote symptomatic versus asymptomatic mandarin leaf samples across seasons; and p755969-lim-vs-or compares finger lime and orange leaves. Full details are provided in Appendix A.

**Table 1 plants-14-01792-t001:** Descriptive statistics of the percentage of reads mapped to the *C. sinensis* DHSO v3.0 and *C. clementina* v1.0 reference genomes using the STAR and HISAT2 alignment tools.

Aligner	Genome	Minimum (%)	Q1 (%)	Median (%)	Q3 (%)	Maximum (%)
STAR	*C. sinensis* DHSO v3.0	23.02	63.53	88.88	92.1	95.71
STAR	*C. clementina* v1.0	27	66.23	89.75	92.41	95.44
HISAT2	*C. sinensis* DHSO v3.0	19.17	56.05	78.06	83.3	88.6
HISAT2	*C. clementina* v1.0	14.73	55.63	80.87	85.52	91.81

## Data Availability

All data is in the public domain.

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
