# Peer review of "Exploring the Genetic Networks of HLB Tolerance in Citrus: Insights Across Species and Tissues"

_plants, 2025, doi:10.3390/plants14121792_

Round 1

Reviewer 1 Report

Comments and Suggestions for Authors

1.      The abstract is currently three paragraphs; most published papers use a single-paragraph abstract.

2.      It's unclear whether normalization was performed across samples, tissues, and studies. The conclusion that HLB's impact on gene expression is subtle or tissue-specific (Figure 2) raises questions about the normalization process.

3.      Given the study's focus on genetic networks of HLB tolerance in citrus, differential expression analysis is crucial. The methods for identifying HLB tolerance genes and accounting for covariates like tissue type need clarification.

Author Response

  1. The abstract is currently three paragraphs; most published papers use a single-paragraph abstract.

We thank for the suggestion. We have reduced and adapted the abstract accordingly to improve readability and comply with journal formatting.

  1. It's unclear whether normalization was performed across samples, tissues, and studies. The conclusion that HLB's impact on gene expression is subtle or tissue-specific (Figure 2) raises questions about the normalization process.

We appreciate this concern and have expanded our explanation of the normalization process in the Methods section . Our workflow includes quality trimming, mapping using STAR, expression quantification with featureCounts, and variance-stabilizing transformation (VST) using DESeq2. Importantly, we applied in this new version ComBat-seq to correct for batch effects arising from differences in sequencing platform and Bioproject origin. Now PCA results  before and after correction illustrate the effective reduction of batch effects. These steps ensure that observed patterns are biologically meaningful and not artifacts of technical variation.

References: Zhang et al. (2020). ComBat-Seq: Batch Effect Adjustment for RNA-Seq Count Data. NAR Genom Bioinform, 2(3), lqaa078. 3.     

  1. Given the study's focus on genetic networks of HLB tolerance in citrus, differential expression analysis is crucial. The methods for identifying HLB tolerance genes and accounting for covariates like tissue type need clarification.

 We clarified in the revised Methods and Results sections that DEGs were identified within gene modules associated with Citrus species, Tissue, and DPI traits. We explicitly defined which bioproject comparisons reflect HLB tolerance or susceptibility. By combining WGCNA module-trait correlations with DEG analysis, we controlled for potential confounding effects from tissue type or developmental stage. This strategy allowed us to identify robust candidate genes linked to HLB tolerance.

Final comments

We would like to sincerely thank the reviewer for raising the issue of batch effects. This observation led to a major re-analysis of our data using ComBat-seq, which had not been previously implemented. As a result, approximately 70–80% of the modules identified changed, significantly altering the structure of the gene co-expression network. Consequently, we revised the main analyses and conclusions accordingly. We also extended the analysis by exploring negatively correlated modules and incorporating moderate correlation thresholds, which enriched the biological interpretation and added depth to the findings. These extensive revisions required considerable time and effort, and we apologize for the delay in submitting this response. The manuscript was edited with 'Track Changes' mode enabled to transparently show all modifications relative to the previous version.

Reviewer 2 Report

Comments and Suggestions for Authors

The manuscript entitled " Exploring the genetic networks of HLB tolerance in citrus: Insights across species and tissues” by Machado et al., employed co-expression analysis of RNA-seq data across multiple citrus species, tissue types and infection states to understand the HLB pathogenesis and resistance mechanisms.

The study was carried out very elaborative but require the following changes before it can be considered for publication in Plants.

The introduction talks about the HLB severity and its economic impact. However, the study objectives could include about how findings will directly facilitate disease management strategies. Moreover, the authors should emphasize how their results make the HLB research more exciting.

In materials and methods section, the authors should expand on the justification provided for the selected parameters for WGCNA.

Regarding, data integration, the authors should discuss about potential batch effects due to differences in experimental conditions and sequencing platforms used. The authors should mention if normalization is enough to take care of these effect.

As authors reported that large number of genes remain unannotated. The authors should mention specific genes that can be good candidates for further functional validation with reasons for their selection. Also, authors should talk about impact of unannotated genes on their findings and future research.

The authors should provide in-depth discussion about pathways that could be linked to HLB tolerance mechanisms. Moreover, the role of ribosomal protein synthesis and secondary metabolism should be detailed.

The authors should validate the key findings through experiments (such as qPCR or knockout/knockdown studies). The authors need to run qPCR for few representative genes at least.

Acronyms (such as DPI) should be defined at first use and used consistently throughout the manuscript.

Overall, the manuscript can be accepted for publication after addressing the above-mentioned critical comments.

Author Response

Comment 1: The introduction talks about the HLB severity and its economic impact. However, the study objectives could include how findings will directly facilitate disease management strategies. Moreover, the authors should emphasize how their results make the HLB research more exciting.

We appreciate this suggestion. In the revised Introduction and Conclusion, we emphasized how our findings provide key molecular targets for breeding and early diagnostics. By revealing tissue- and genotype-specific defense responses, the study offers novel avenues for citrus improvement and positions transcriptomics as a valuable tool in HLB research.

Comment 2: In the Materials and Methods section, the authors should expand on the justification provided for the selected parameters for WGCNA.

 According to the reviewer suggestion, we expanded the Materials and Methods section to justify the selection of WGCNA parameters. The soft-thresholding power (β = 7) was chosen based on the scale-free topology criterion (R² ≥ 0.90). The minimum module size (30) and merge cut height (0.25) were selected to balance sensitivity and module robustness, following WGCNA best practices.

Comment 3: Regarding data integration, the authors should discuss potential batch effects due to differences in experimental conditions and sequencing platforms used. The authors should mention if normalization is enough to take care of these effects.

We acknowledge this issue and have addressed it in both the Methods and Results sections. We applied ComBat-seq for batch correction after VST normalization to account for differences among the 17 integrated bioprojects. PCA results confirmed that this approach effectively minimized non-biological variation.

References: Zhang et al. (2020). ComBat-Seq: Batch Effect Adjustment for RNA-Seq Count Data. NAR Genom Bioinform, 2(3), lqaa078. 3.     

Comment 4: As authors reported that a large number of genes remain unannotated. The authors should mention specific genes that can be good candidates for further functional validation with reasons for their selection. Also, authors should talk about the impact of unannotated genes on their findings and future research.

In the new manuscript version, we highlight specific unannotated genes (e.g., Cs_ont_7g002890.1 and LOC102612408) with strong module-trait correlations and consistent upregulation in tolerant genotypes or tissues. These genes are promising candidates for functional validation due to their predicted roles in stress response and cellular organization. We also discuss the limitations and opportunities posed by unannotated genes in the Discussion section.

Comment 5: The authors should provide in-depth discussion about pathways that could be linked to HLB tolerance mechanisms. Moreover, the role of ribosomal protein synthesis and secondary metabolism should be detailed.

We appreciate this suggestion. We expanded the Discussion to include detailed interpretation of modules enriched in ribosomal protein synthesis and secondary metabolism pathways. These functions may support rapid protein turnover and oxidative stress mitigation in tolerant citrus genotypes.

Comment 6: The authors should validate the key findings through experiments (such as qPCR or knockout/knockdown studies). The authors need to run qPCR for few representative genes at least.

We agree that experimental validation is critical. However, the focus of this manuscript is based on in-silico analysis and computational integration. Currently we are preforming our own dataset with a new plant infection experiment in control conditions. This work is in progress and we plan to design qPCR assays for representative genes from key modules (e.g., trehalose metabolism, plastid function). These validations will be part of a future study.

Comment 7: Acronyms (such as DPI) should be defined at first use and used consistently throughout the manuscript.

We have revised the manuscript to ensure that all acronyms, including DPI (Days Post-Infection), are defined at first mention and used consistently thereafter.

Final comments

We would like to sincerely thank the reviewer for raising the issue of batch effects. This observation led to a major re-analysis of our data using ComBat-seq, which had not been previously implemented. As a result, approximately 70–80% of the modules identified changed, significantly altering the structure of the gene co-expression network. Consequently, we revised the main analyses and conclusions accordingly. We also extended the analysis by exploring negatively correlated modules and incorporating moderate correlation thresholds, which enriched the biological interpretation and added depth to the findings. These extensive revisions required considerable time and effort, and we apologize for the delay in submitting this response. The manuscript was edited with 'Track Changes' mode enabled to transparently show all modifications relative to the previous version.

Round 2

Reviewer 1 Report

Comments and Suggestions for Authors

No more comments.

Reviewer 2 Report

Comments and Suggestions for Authors

The replies to the comments are satisfactory and the revised manuscript can be accepted for publication.